

# Enhancing the Lagrangian approach for moisture source identification through sensitivity testing of assumptions using BTrIMS1.1

Yinglin Mu[1,3], Jason P Evans[1,3], Andrea S. Taschetto[1,3], Chiara Holgate[2,3]

[1] Climate Change Research Centre, University of New South Wales, Sydney, 2052, Australia

[2]  Research School of Earth Sciences, Australia National University, 0200, Canberra

[3] ARC Centre of Excellence for 21st Century Weather

*Correspondence to*: Yinglin Mu (yinglin@unsw.edu.au)

**Abstract.** Moisture is the fundamental basis for precipitation, and understanding the sources of moisture is crucial for comprehending changes in precipitation patterns. Lagrangian models have been employed for moisture tracking in both
extreme weather events and climatological studies as a means to gain insight into driving physical processes. Lagrangian moisture tracking models follow independent air parcels based on a set of defined assumptions. Despite the existence of many Lagrangian models and studies applying them for moisture tracking, these assumptions are seldom thoroughly tested. In this study, we use the Lagrangian model BTrIMS to demonstrate the impact of these assumptions on the results of moisture source identification. In particular, we test the method's dependence on the number of parcels released; the height
that parcels are released; the vertical movement of air parcels; the vertical well-mixed assumptions that lead to different moisture identification methods along trajectories, the within-grid interpolation method and the back-trajectory time step. We find that releasing approximately 200 air parcels per day from each grid point is necessary to obtain accurate results for a region of 10 grid points or more (an area of ~9,000km$^2$ in this case). Additionally, the vertical movement of air parcels, their release height, and along-trajectory identification method of moisture substantially affect the identified moisture sources,
whereas within-grid interpolation and back-trajectory time step within a reasonable range has a relatively minor role on the results. The mechanisms behind these assumptions involve heat exchange, precipitation formation height, vertical mixing of surface evapotranspiration, and numerical noise, all of which must be carefully considered for realistic results.
Based on the results of sensitivity tests and analysis of underlying mechanisms behind the assumptions, we improve the Lagrangian model BTrIMS1.0 to a new version (BTrIMS1.1) for broader applicability. The findings of this study provide
critical information for improving Lagrangian moisture source identification methods in general and will benefit future research in this field, including studies examining changes in moisture sources due to climate change.
**Key words:** precipitation, moisture source, back-trajectory methods, Lagrangian, model assumption, air parcels



## 1 Introduction

Moisture is a prerequisite for precipitation. Moisture sources refer to the origins of moisture, given by evapotranspiration
from land or ocean surfaces (Gimeno et al., 2012). Identifying moisture sources for given precipitation events or climatology
has generated significant curiosity in atmospheric science and hydrology, though challenges remain, including the difficulty
in validating results.

Early studies focus on estimating precipitation recycling ratios from global datasets, which represent the contribution of local
evapotranspiration (ET) to local precipitation, thereby measuring land surface-atmosphere interactions. Other studies use 1D
(Budyko, 1974) and 2D analytical models (Brubaker et al., 1993; Dominguez et al., 2020; Eltahir and Bras, 1996). Generally
speaking, the simpler the model is, the more assumptions it involves, which can compromise the accuracy of the results. One
common assumption is that ET from the surface is vertically well-mixed in the atmosphere (Eltahir and Bras, 1996).
Analytical methods rely on the atmospheric water mass conservation equation (also known as 'water budget equation') to
estimate regional recycling ratios, but these methods oversimplify reality by assuming uniform moisture mixing. Another
limitation of precipitation recycling approaches is its dependence on both the spatial scale and shape of the study region,
which makes direct comparison across different regions challenging. However, some methods and formulations have been
proposed to obtain metrics that are independent of the shape and scale of study regions (van der Ent and Savenije, 2011).

More recent studies estimate moisture sources through one of three main approaches: offline Eulerian methods, online
Eulerian methods, and Lagrangian methods.

Offline Eulerian methods use already existing atmospheric data as input (Goessling and Reick, 2013), and are run on
Eulerian grids, like the Water Accounting Model (WAM) (van der Ent et al., 2010). Some methods obtain the distribution of
continental moisture recycling ratios (Goessling and Reick, 2011), while others determine the moisture source distribution
by applying water mass conservation among grids (van der Ent et al., 2013; van der Ent and Savenije, 2011). These models
are less accurate in regions with significant vertical shear but have been improved by incorporating an additional vertical
layer (van der Ent et al., 2013).

Online Eulerian methods track moisture by tagging it upon entering the model atmosphere, either through surface ET or
from the lateral boundaries, and following its journey through processes like precipitation and mixing. The tracking process
is incorporated into the model in parallel with the water accounting process. Examples of online Eulerian methods include
COSMO model (Winschall et al., 2014) and WRF-WVT (Dominguez et al., 2016). The former is primarily used for
climatological studies, while the latter is more frequently employed for regional research requiring higher resolution. While
these models are precise, they require predefined water source regions and are limited to forward tracking.

Lagrangian methods can track moisture either forward or backward in time and space. For the purpose of tracking moisture
sources for precipitation, only backward Lagrangian methods are discussed herein and are referred to as "Lagrangian
methods". These methods track moisture sources by calculating the trajectories of independent air parcels and
conceptualizing the physical processes involved. The wind field is typically used for advection of air parcels in trajectory




calculations, and moisture sources are identified along these trajectories. To clarify, there are two important sub-processes of Lagrangian methods for moisture tracking: calculation of trajectories and identification of moisture sources along trajectories.

Models for calculating backward trajectories include LAGRANTO (Sprenger and Wernli, 2015), HYSPLIT (Draxler and
Hess, 1997), FLEXTRA (Stohl et al., 1995), FLEXPART (Pisso et al., 2019), Utrack (Tuinenburg and Staal, 2020), and BTrIMS (Holgate et al, 2020). Additionally, there are software and packages that enhance the usability of these models (Fernández-Alvarez et al., 2022; Pérez-Alarcón et al., 2024). These Lagrangian models numerically solve the air parcel trajectory equation(Eq.(1) below) but differ in the number of iterations, treatment of turbulence, interpolation method, time step and output format:

$\quad \frac{Dx}{Dt} = u(x)$ (1)

$Dx$: air parcel's displacement in one time step;

$Dt$: time step;

$u(x)$: velocity field

For calculation of trajectories, most Lagrangian models use wind fields to drive the horizontal and vertical movement of air
parcels. However, Dirmeyer and Brubaker (1999) suggested that air parcels can also move quasi-isentropically due to sufficient stratification above the planetary boundary layer (PBL). In a convectively heated boundary layer, the potential temperature of an air parcel can increase, which manifests as a potential temperature reduction in backtracking. While turbulence affects atmospheric conditions, it is generally not accounted for in moisture back-tracking due to its random nature, leading to no net vertical drift. However, some models, like HYSPLIT, do consider this factor.

Two main methods are used for identifying moisture sources with Lagrangian trajectory models. The first, assumes that ET is vertically well-mixed throughout the atmospheric column. This means that moisture from ET is directly related to moisture in air parcels at any height in the atmospheric column and the percent of moisture from the current grid point where the air parcel resides is the same throughout the whole column. This identification method is adopted by Dirmeyer and Brubaker (1999), Utrack (Tuinenburg and Staal, 2020) and BTrIMS (Holgate et al., 2020). For brevity, we will hereafter
refer to this method as 'ET-mixing'. The second method diagnoses moisture gain and loss from changes in specific humidity along an air parcel's trajectory. Unlike the first method, it doesn't rely on the assumption that ET is vertically well-mixed in the atmospheric column, nor does it explicitly relate air parcel moisture to surface ET. However, if interpreting the result gained from this method as moisture sources, it is implicitly assumed that increase of specific humidity is solely caused by surface ET. This second method, known as 'WaterSip', was proposed by Sodemann (2008) and has since been applied in
other studies (Brunello et al., 2024; Fremme and Sodemann, 2019a, b; Hu et al., 2021; Keune et al., 2022; Terpstra et al., 2021; Zhang et al., 2022). Generally, only a part of moisture in an air column transforms to precipitation, so the vertical distribution of precipitation formation is vital in tracking moisture sources and these two identification methods also differs





in this: ET-mixing distributes the precipitation formation proportionally by moisture content, while WaterSip sets threshold for relative humidity and specific humidity decreases.

Due to limitations in Eulerian methods for moisture tracking, Lagrangian methods are widely used. However, their underlying assumptions introduce uncertainties that are rarely evaluated systematically. Existing studies examined some assumptions, and these are often restricted to specific precipitation types or regions (Crespo-Otero et al., 2024; Pérez-Alarcón et al., 2025; Tuinenburg and Staal, 2020).

In this study, we aim to assess the sensitivity of Lagrangian moisture source identification to various assumptions by
applying different choices for the same assumption. The sensitivity tests include:

**Number of air parcels to release:** Releasing more air parcels generally improves the precision of the results, but also increases computational cost. Therefore, the optimal number of air parcels will produce an accurate identification of moisture sources with minimal computational cost.

**Vertical distribution of precipitation formation:** The altitude at which precipitation forms determines which portion of the
atmospheric moisture column should be tracked — a choice that is operationalized through the selection of air parcel release height.

**Vertical movement of air parcels:** The assumption of air parcel's vertical movement determines the trajectory and subsequent moisture identification and there are three schemes based on different theories.

**Vertical well-mixed assumption:** This assumption posits that ET from the surface is well-mixed within the boundary layer
or the entire atmospheric column, thereby linking ET along the trajectory to resulting precipitation. Accepting or avoiding this assumption leads to different moisture identification methods along air parcel trajectories.

**Linear change of variables within grids:** Physical variables are typically interpolated linearly within grid cells to the positions of air parcels. However, cubic interpolation methods have also been employed in some Lagrangian studies, as they are thought to better capture sub-grid scale variability.

**Trajectory time step:** Trajectory time step has a clear influence on path calculated. Moreover, under the vertical well-mixed assumption, there is implicit requirement that ET becomes well-mixed within one trajectory time step, making the choice of time step particularly important for moisture source identification.

According to the subprocesses of backward Lagrangian methods, the sensitivity tests of assumptions in this study aim to answer the following questions, and section 2.4 is organized accordingly:

(1) How many air parcels should be released to obtain realistic results?

    (2) How should the air parcels be vertically distributed upon release?

    (3) How should an air parcel move in the vertical?

    (4) How does surface ET mix in the air column under different conditions?

    (5) How does the interpolation method within grid influence the result?

(6) How will the back-trajectory time step influence the results?



## 2 Methods and Data

### 2.1 Model

The model employed for sensitivity tests in this study is the Back-Trajectory for the Identification of Moisture Sources (BTrIMS). This model, originally developed and applied to investigate precipitation recycling and evaporative moisture
source regions in Australia (Holgate et al., 2020), releases a fixed number of air parcels at each grid point where daily precipitation exceeds a threshold (1 mm in this study). The air parcels are advected by wind within a predefined large domain surrounding the study region, with their motions treated as independent. Details regarding the release time and height (vertical position) are provided in sections 2.4.1 and 2.4.3. Backward trajectories are calculated using a fully implicit integration method by Merrill (1986) horizontally, with choices for vertical movement described in section 2.4.2.
For each air parcel, moisture contributions to precipitation are identified at grid points along the trajectories until all (over a given threshold of percentage) moisture for the precipitation is found, the backtracking period ends, or the parcel exits the model domain. The backtracking period can vary across different regions and events due to spatial and temporal differences in multiple processes such as evapotranspiration advection, turbulent mixing, precipitation, and others. Nevertheless, a period of 20 days is generally sufficient under various definitions of water vapor residence time (Gimeno et al., 2021).
Therefore, in this study, the backtracking period is set to 20 days, and we find that nearly all moisture can be identified for the three events considered. Moisture identification is the process of assigning 100% of the moisture to points along the trajectories (see details in section 2.4.4). In BTrIMS, if an air parcel reaches the boundary of the tracking domain before all moisture sources has been identified, the remaining moisture is assigned to the boundary grid.

### 2.2 Data

In this study, ERA5 reanalysis data (Hersbach et al., 2020) are used as input. Different variables are used depending on the setup of each experiment. For pressure-level variables, 37 pressure levels are available, ranging from 1000hPa to 1hPa. The pressure level variables used are listed below:

The 3D wind fields (U, V, W) are used to calculate back trajectories. The specific content of water vapor (q) and condensed hydrometeors, including cloud liquid water (CLWC), cloud rain (CRWC), cloud ice (CIWC), and cloud snow (CSWC) are
used for water content in air parcels and column precipitable water (PW). Temperature (T) is used to calculate potential temperature ($\theta$) and equivalent potential temperature ($\theta_e$).

For single-level variables, evapotranspiration (ET), total precipitation (TP), and convective precipitation (CP) are used in moisture identification. Surface pressure (SP) is used to derive the vertical precipitable water profile and to determine whether an air parcel has reached the land or ocean surface, enabling adjustments when surface contact occurs. The
boundary layer height (BLH) is used when the entire atmospheric column needs to be divided into planetary boundary layer (PBL) and the free atmosphere above it.





ERA5 hourly precipitation data represent accumulated precipitation over the preceding hour. When the model time step is smaller than one hour (e.g.,15 minutes), the precipitation is evenly distributed across sub-hourly intervals. Other variables, such as evapotranspiration (ET), are also linearly interpolated in time. In the model, precipitation occurring at a given time

step is assumed to be caused only by ET prior to that time, reflecting the physical lag between ET and precipitation formation.

## 2.3 Three precipitation events

An assumption that is physically meaningful in one region or specific type of precipitation events may deviate significantly from reality in another region or under different meteorological conditions. To assess the regional sensitivity of these

assumptions, this study examines three distinct precipitation events: the February 2022 event in Australia, the August 2022 event in Pakistan, and the October 2023 event in Scotland.

### 2.3.1 February 2022 Australia precipitation event

Rainfall along Australia's east coast is influenced by several modes of climate variability, including the El Niño–Southern Oscillation (ENSO), Indian Ocean Dipole (IOD), Southern Annular Mode (SAM), and the Madden–Julian Oscillation

(MJO). Interactions among these modes can enhance or suppress the others' effects, shaping the spatial and temporal variations of precipitation and extreme events in Australia. Other key drivers include variations in the position and strength of subtropical ridge; land-atmosphere feedbacks; synoptic-scale processes such as Rossby wave breaking; and weather patterns like tropical and extratropical cyclones, blocking highs, and atmospheric rivers (Barnes et al., 2023; De Vries et al., 2023; Pepler et al., 2019; Reid et al., 2022, 2025; Sekizawa et al., 2023).

The year 2022 saw particularly heavy rainfall in Australia, with several record-breaking events. One such event occurred during February-March, characterized by persistent heavy rainfall (Dao et al., 2023). February 25[th] was chosen for moisture tracking during this period of intense rainfall. The study region spans -22N-32N and 149E-158E (outlined in pale yellow in Fig.3), covering the east coast of Australia, which experienced significant rainfall during February and March.

During this event, La Niña and positive SAM were active, promoting rainfall, while the IOD remained inactive. The MJO

was also inactive for most of the period (Reid et al., 2025). According to Barnes et al (2023), Rossby-wave breaking developed a slow-moving coherent cyclonic potential vorticity anomaly, causing a series of cyclones to stall in the region, which ultimately generated extreme rainfall anomalies across east Australia during this event. Rossby wave breaking, in combination with moisture transport, has been estimated to contribute to 70% of all extreme events in subtropical and extratropical regions including in eastern Australia (De Vries, 2020).




### 2.3.2 Pakistan and Scotland precipitation events

For simplicity, we focus our study on the Australian event. However, we also tested the model assumptions against two other precipitation events that incurred significant impacts on local communities, namely the Pakistan floods in August 2022 (Chen et al., 2024) and Scotland rainfall in October 2023 (Graham et al., 2023). The synoptic conditions of the Pakistan and

Scotland rainfall events are described in the Supplementary material.

### 2.4 Sensitivity experiments

### 2.4.1 Number of air parcels to release

Similar to other Lagrangian models, BTrIMS identifies surface moisture sources by releasing a number of air parcels each day at each grid point in the study region, with each air parcel representing an equal fraction of precipitation. The air parcels

are launched at a rate proportional to the temporal rate of precipitation and are released from a random, total-precipitable-water-weighted height in the atmosphere. In the model, these two processes are implemented stochastically. Releasing more air parcels enhances the precision of resultant moisture sources but also increases the required computing resources. To evaluate the sensitivity of the results to the number of parcels released, we assess the relationship between the number of air parcels released daily per grid point and the precision of the results. Precision is quantified as the pattern correlation between

the results of different parcel release numbers and the 'truth'. Here, the result of 1000 air parcels released daily at each grid point is considered as the 'truth'. The sensitivity test includes various numbers of air parcels released: 50, 100, 200, and 500, with results presented in section 3.1.

### 2.4.2 Vertical movement of air parcels

The stepwise movement of individual air parcels in the atmosphere is fundamental in trajectory calculations as each step

cumulatively builds the parcel's path and influences subsequent moisture uptake and loss, ultimately affecting the identification of moisture sources. In all existing Lagrangian models, the horizontal advection of air parcels is driven by the horizontal wind field. However, for vertical motion, there are primarily two different assumptions. The first assumes that vertical motion, like horizontal advection, is forced by the vertical wind field. The second assumption, proposed by Dirmeyer & Brubaker (1999), postulates that air parcels move quasi-isentropically in the vertical, assuming sufficient

stratification outside the boundary layer. Within the boundary layer, potential temperature is modified by convective diabatic processes. In this study, we also evaluate a third assumption: the air parcels move vertically along the equivalent potential temperature ($\theta_e$), which remains conserved in both dry and moist adiabatic processes. The latter two schemes incorporate thermodynamic theories to approximate the movement of air parcels, which differs from the approach solely based on wind fields.





In all three schemes, parcels that descend below the surface pressure are repositioned to 5 hPa above the surface pressure to
avoid unrealistic behavior. The equivalent potential temperature is calculated using the formula from Emanuel (1994) as
follows.

$$es = 611 \exp\left(\frac{17.2964 \times (T - 273.16)}{T - 35.86}\right)$$

$$mix_s = \frac{R_d}{R_v} \times \frac{es}{P - es}$$

$$\theta_e = T \times \frac{P_0}{P}^{\frac{Rd}{Cp + C_l \times mixtot}} \times mix_s^{(-mix \times \frac{R_v}{Cp + C_l \times mix_{tot}})} \times \exp\left(L_v \times mix/(C_p + C_l \times mix_{tot}) \times T\right) \tag{2}$$

$es$: saturation vapor pressure,Pa;

$T$: air temperature, K;

$mix_s$: saturated mixing ratio, kg/kg;

$P$: air pressure, Pa;

$P_0$: reference surface pressure, 1000hPa;

$C_p$: heat capacity of air at constant pressure, 1004.67 (J/kgK);

$C_l$: heat capacity of liquid water at ~-20C, 4400 (J/kgK);

$mix$: specific humidity of water vapor, kg/kg;

$mix_{tot}$: specific humidity of total water, kg/kg;

$L_v$: latent heat of vaporization of water, 2.25E6 (Jkg-1);

$R_d$: ideal gas constant for dry air, 287.053 (J/kgK);

$R_v$: gas constant of water vapor, 461.5 (J/kgK).

### 2.4.3 Parcel release height

Parcel release height refers to the initial vertical position of air parcels for backward trajectory calculation. In BTrIMS, air
parcels are released randomly based on the humidity profile in a humidity-weighted manner, under the assumption that the
vertical distribution of precipitable water indicates the probability of precipitation formation. This approach is similar to that
of Dirmeyer and Brubaker (1999) but differs from Sodemann (2008). Different atmospheric water species can be included
into the humidity profile to account for the precipitation formation process. While past studies have typically considered only

water vapor, cloud hydrometeors also contribute to the atmospheric water mass and can even be more important, as cloud
formation is a prerequisite for precipitation. Although water vapor constitutes the majority of atmospheric precipitable water,
neglecting hydrometeors could result in an incomplete representation of the vertical moisture distribution.

To assess how the inclusion of different water species affects the humidity profile, thereby influencing the initial vertical
position of air parcels and ultimately the identification of moisture sources, this study tests three configurations for the





vertical distribution of humidity: (1) all water species (both water vapor and hydrometeors), (2) water vapor only, and (3) cloud hydrometeors only.

### 2.4.4 Vertically well-mixed assumption and moisture identification along trajectories

The vertically well-mixed assumption posits that the ET from the land or ocean surface is vertically well-mixed throughout the entire atmospheric column or within the planetary boundary layer (PBL) at the temporal resolution of the back trajectory
time step. Under this assumption, at some time step, the ET entering the atmosphere is proportionally mixed, such that the ratio of ET to total precipitable water (ET/TPW) represents the fraction of moisture contributed by this grid point to an individual air parcel at that time step. This assumption was first introduced for moisture tracking by Dirmeyer & Brubaker (1999), and is referred to in this study as the 'ET-mixing' method to reflect the underlying physical mechanism.

An alternative moisture identification approach, WaterSip, avoids the well-mixed assumption, by proposing that moisture
uptake and loss is indicated by changes in an air parcel's specific humidity. One challenge with WaterSip is that change in specific humidity result from both evapotranspiration (ET) and precipitation (P) occurring simultaneously, making it difficult to distinguish the two based solely on specific humidity increase. However, Stohl & James (2004, 2005) demonstrated that within a given time interval, only one of them, either ET or P, tends to dominate, with the other being negligible. Furthermore, changes in specific humidity can also be related to atmospheric convergence or divergence processes, rather
than local ET or P. In addition to the basic principle, WaterSip impose additional criteria to determine whether moisture uptake is actually occurring:

(1) The specific humidity increase must exceed a threshold, typically set to $\Delta q = 0.05$ g kg$^{-1}$,

(2) Moisture uptake must occur within the PBL, as surface ET does not contribute to moisture increases above PBL.

(3) To filter trajectories that contribute to final precipitation, relative humidity must exceed 80%, as in the lower troposphere,
specific humidity of air parcels can be high, but the air may not be saturated (i.e., low relative humidity).

These assumptions formed the basis of WaterSip method, which was first proposed by Sodemann (2008) based on the work of Stohl & James (2004, 2005). Subsequent studies suggest that surface ET can also influence moisture above the PBL through convection (Fremme and Sodemann, 2019b). Therefore criteria (2) is not required.

In this test, the two methods are tested in the entire atmospheric column as Option 1 (ET-mixing method) and Option 2
(WaterSip method). Compared to Sodemann (2008), Option 2 retains the threshold for specific humidity increase, $\Delta q = 0.05$ g kg$^{-1}$, but does not impose a threshold on initial relative humidity or require moisture uptake to occur within the PBL. The rationale for omitting the relative humidity threshold is that subgrid processes can allow the lower troposphere to contribute to precipitation.

Atmospheric theory widely supports the assumption of vertical well-mixing within the PBL; however, above the PBL,
stratification is strong and mixing is limited. Therefore, Option 3 introduces a hybrid approach by dividing the air column into the PBL and the free troposphere. Within the PBL, the ET-mixing identification method is applied, and the contribution ratio is ET/TPW$_{pbl}$, while above the PBL, the WaterSip method is used.





When convection occurs, the mixing characteristics of the air column change. In the case of deep convection, the entire air column can become well-mixed, allowing surface ET to contribute to moisture above PBL. To account for this, Option 4

assumes that when convective precipitation exceeds 0.1 mm, the entire air column is well-mixed, and the ET-mixing method is applied for moisture identification at this grid point.

Finally, 'older' moisture within air parcels is gradually diluted by newly acquired moisture along the trajectory. This dilution effect necessitates the use of a weighting factor in both methods ($fac_{ET}$ for ET-mixing and $fac_q$ for WaterSip).

The procedures of Option 1 and Option 2 are illustrated below in Eq. (3) and Eq. (4) and adjustments are made for Option 3

and Option 4 accordingly as described above:

'ET-mixing' method (Option 1):

$$wv_{cont} = 0.$$

$$fac_{wv} = 1.$$

$$frac_{n1}' = \frac{ET_n}{TPW_n}$$

$$frac_{n1} = frac_{n1}' \times fac_{wv} \tag{3}$$

$$fac_{wv} = fac_{wv} \times (1 - \frac{ET_n}{TPW_n})$$

$$wv\_cont = wv\_cont + frac_{n1}$$

WaterSip method (Option 2):

$$wv\_cont = 0.$$

$$fac_q = 1.$$

$$frac_{n2}' = \frac{q_n - q_{n-1}}{q_n}, (q_n - q_{n-1} > 0.05 g/kg)$$

$$frac_{n2} = frac_{n2}' \times fac_q \tag{4}$$

$$fac_q = fac_q \times (1 - \frac{q_n - q_{n-1}}{q_n})$$

$$wv\_cont = wv\_cont + frac_{n2}$$

### 2.4.5 Within grid interpolation

Along its trajectory, an air parcel is not necessarily located at the center of a grid cell, requiring interpolation of variables such as wind and specific humidity. Stohl et al. (2001) and Tuinenburg and Staal (2020) demonstrated that switching horizontal interpolation from bilinear to bicubic, or to nearest neighbour, respectively, altered estimated parcel trajectories to

some extent. To investigate the sensitivity of moisture sources to the horizontal interpolation method with ERA5 reanalysis as forcing data, the performance of bilinear and bicubic within-grid interpolation methods is compared in this test.



### 2.4.6 Back trajectory time step

The trajectory of an air parcel evolves step by step, with each point along a parcel's trajectory influenced by the previous one. The choice of time step is crucial, as it directly impacts the precision of the trajectory calculations, as well as other
variables, such as wind and specific humidity, which in turn affect the overall moisture tracking results. A smaller time step enhances precision but increases computational cost, whereas a larger time step sacrifices precision while reducing computational expense. Finding an optimal time step is essential to balance precision and efficiency. In this study, four different time step intervals are evaluated: 10, 15, 30 and 60 minutes.

### 3 Results and discussion

**3.1 Number of air parcels to release**

For each precipitation event, grid points with daily precipitation over 1mm within the study region are selected and pattern correlations are calculated between the results of 50, 100, 200, and 500 air parcels per grid per day and the 'truth' (1000 air parcels), respectively. Figure 1 presents the results for the Australian event, Fig. S1 and Fig. S3 for the Pakistan and the Scotland event, with panels (a~c) illustrating different vertical movement schemes.

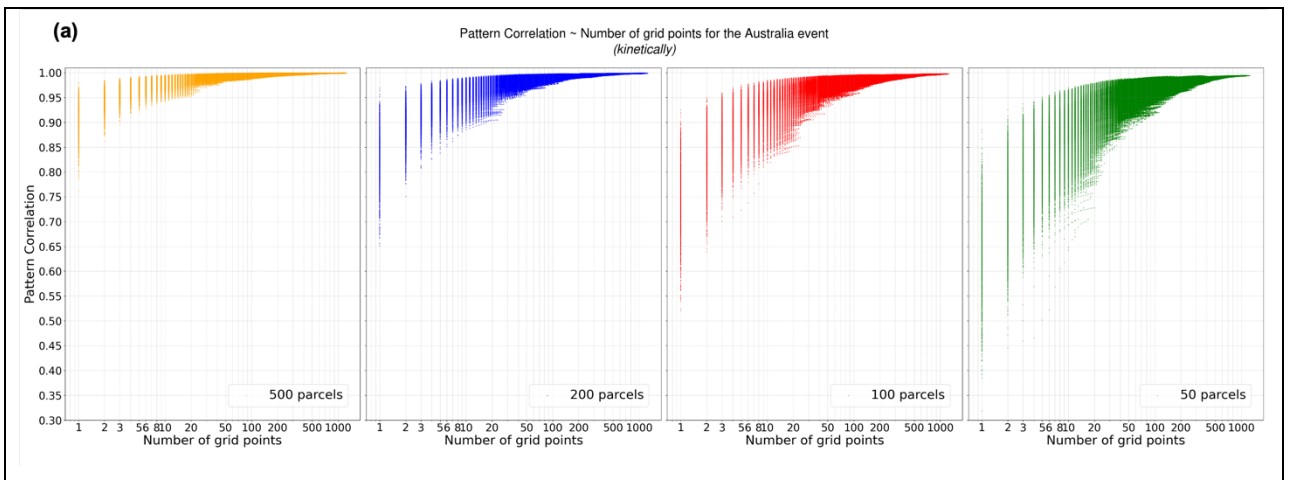



**Figure 1 Sensitivity of modelled moisture sources to the number of parcels released and vertical movement schemes for the Australia event. Pattern correlation of moisture source (y-axis) is plotted against the number of grid points (x-axis). Each panel presents results for different numbers of air parcels released per day: 500 parcels (yellow), 200 parcels (blue), 100 parcels (red), and 50 parcels (green). Each row corresponds to a different vertical movement scheme: (a) kinetic, (b) isentropic, and (c) along equivalent potential temperature.**

The results of three air parcel vertical movement schemes are similar across all three events: the more air parcels released each day, the higher the pattern correlation, indicating increasing precision (Fig. 1; Fig. S1 and Fig. S3). Additionally, increasing the number of grid points (i.e. increasing the spatial area) improves the precision of the results. However, a minimum threshold for pattern correlation of each individual grid point must also be considered. Based on the pattern correlation results presented in Fig. 1, Fig. 2 applies three threshold values (0.95, 0.90, and 0.85) to summarize the relationship between the number of grid points and the number of air parcels required to achieve the threshold value for the Australian event (similarly, the Pakistan and Scotland events can be seen in Fig. S2 and Fig. S4).





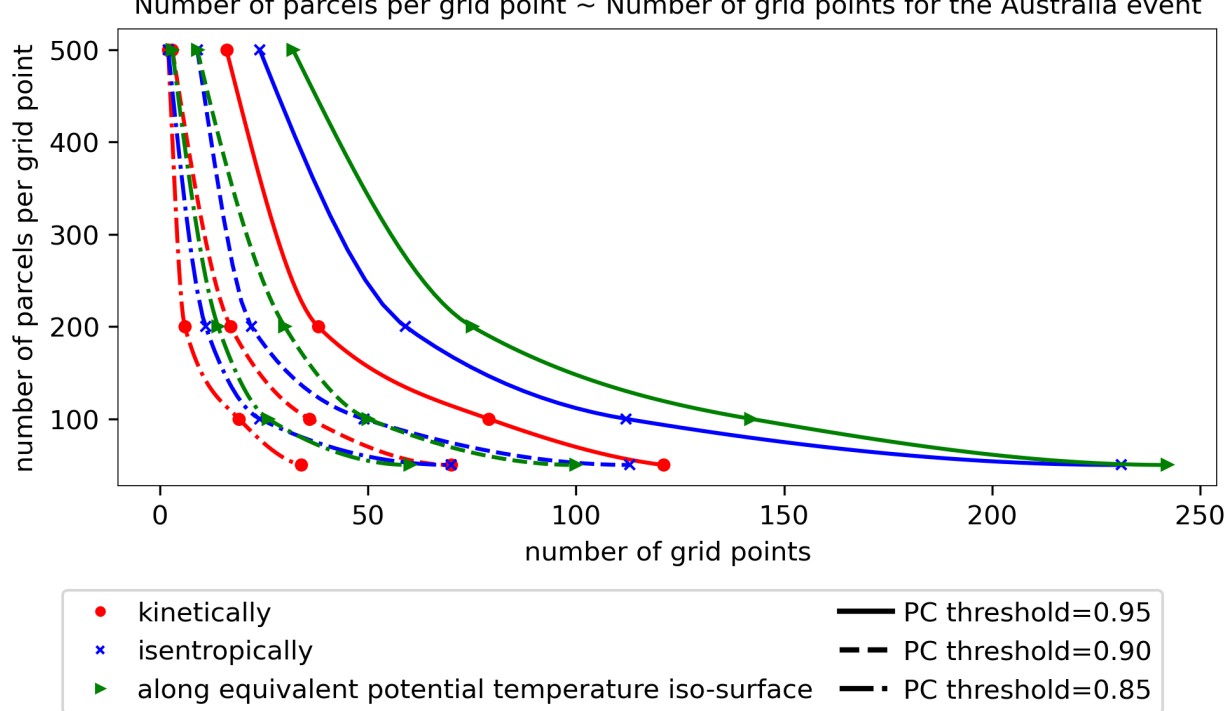

**Figure 2 The relationship between the number of air parcels and the number of grid points required to achieve a pattern correlation of 0.95, 0.90 and 0.85, dot, cross and triangle for kinetic, isentropic and along equivalent potential temperature schemes. PC means pattern correlation.**

Figure 1 and Figure 2 show that, for the Australia event, using a 3-dimensional wind field to drive air parcels' movement requires fewer air parcels to achieve a stable result for the same number of grid points than the alternative vertical movement schemes tested. For example, for a region with 150 grid points, 100 air parcels per grid point is enough to achieve a pattern correlation of 0.95 with the truth if the kinetic scheme is applied, but not for the $\theta_e$ based scheme. In contrast, for the Scotland and Pakistan events, more air parcels are required when kinetic scheme is chosen compared to the other alternative schemes (Fig. S1-Fig. S4). This difference may be due to variations in the dynamic and thermodynamic conditions of the air. As an example, convection plays an important role in the Australia event, causing great variations in $\theta_e$ field, more air parcels are required using the $\theta_e$ based scheme than the kinetic scheme. The variations of these variables reflect the characteristics of each precipitation event and governing mechanisms. Note that the curves in Fig. 2, Fig. S2, and Fig. S4 are intended as qualitative illustrations only, aimed at showing the relative differences among the various vertical movement schemes of air parcels, since we only tested four air parcels per grid point.

Finally, considering the combined results from all three precipitation events and accounting for both pattern correlation for single grid point and regions with tens of grid points (typical size of an upper reach watershed, ~9,000km²), 200 parcels per day per grid point is identified as the optimal choice, which strikes a balance between ensuring reliable results and optimizing computational resources. Tuinenburg and Staal (2020) also examined the effect of the number of air parcels on



tracking performance, using the continental recycling ratio (CRR) and the mean absolute latitudinal or longitudinal distance as evaluation metrics. They recommended releasing 500 air parcels per millimetre of evaporation; however, this recommendation is based on a single evaporation source over a one-month period to find the downwind precipitation footprints of single evaporation source. They further argue that, for a larger area or a longer study period, less air parcels

may be justified.  Actually, there was little difference in CRR and about 0.1 degree mean absolute distance when 50 or 100/mm air parcels are released in their study.

## 3.2 Vertical movement of air parcels

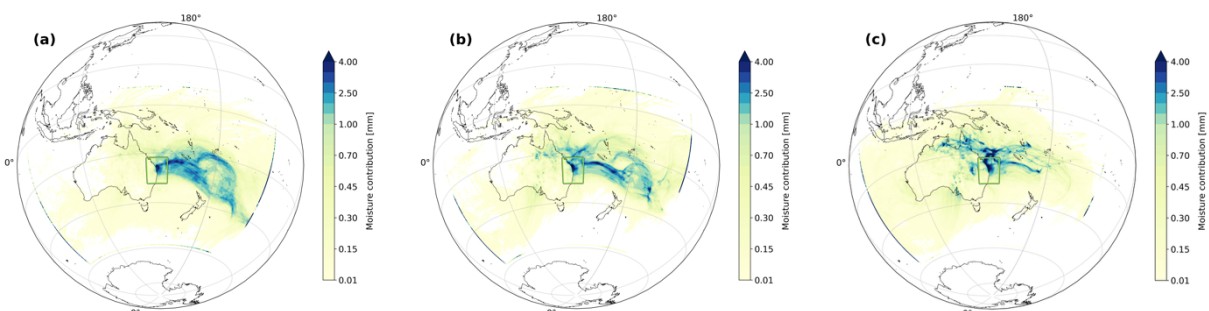

**Figure 3 The moisture source results of using different schemes in air parcel's vertical movement: (a) kinetic, (b) isentropic, and (c) equivalent potential temperature.**

The choice of vertical movement scheme significantly affects the identification of moisture sources for the Australia event (Fig. 3). Using the kinetic scheme, the majority of moisture originates from the South Pacific Ocean, particularly from the subtropics between 20S-40S and extends far eastward in longitude with minimal contribution from land areas and from the west of the study region (Fig. 3a). The isentropic scheme yields relatively similar results (Fig. 3b), but with more zonally-confined oceanic moisture sources and greater moisture contributions from the land areas of northeast Australia. In contrast,

the equivalent potential temperature-based scheme exhibits markedly different moisture sources, with sources confined to the northeast coast of Australia and the Coral Sea (Fig. 3c). Notably, there is also increased moisture contribution from inland regions and from the area south of Australia. Another difference lies in the presence of a well-developed cyclonic boundary within the main moisture source regions under the kinetic scheme, both to the north and south. This boundary becomes less distinct under the isentropic scheme and appears even more diffuse under equivalent potential temperature

scheme.

For the Pakistan event,  moisture sources using the kinetic scheme indicate that a large amount of moisture is transported from the Arabian Sea by the Southeast Asian monsoon system (Fig. S5a). Under the isentropic scheme, significantly more moisture originates from the adjacent southern Himalayas, with reduced contribution from the southern Arabian Sea (Fig. S5b). The result under the equivalent potential temperature scheme differs substantially from that under kinetic scheme, with

the major moisture source shifting to the eastern land region, mainly over China, and the Himalayas having relatively little effect on moisture transport (Fig. S5c).





For the Scotland event, there is some similarity between the results under kinetic and isentropic schemes. Specifically, two major moisture source regions — one over the North Atlantic near the Mediterranean and another over the North Sea — are found under both schemes (Fig. S6a,b). However, under the equivalent potential temperature scheme, relatively more
moisture is sourced from the distant western Pacific Ocean and a region of the North Atlantic Ocean, located west of West Africa. The result obtained using the isentropic scheme appears to fall between those produced by the other two schemes in terms of the identified major source regions.

The assumptions of purely isentropic motion and motion along the equivalent potential temperature ($\theta_e$) iso-surface are not entirely realistic, as both neglect diabatic heating and cooling processes. The isentropic scheme is only valid for dry adiabatic
processes; whereas the $\theta_e$-based scheme assumes that all latent heat released from water vapor condensation is converted into sensible heat, resulting in a temperature increase. Both assumptions represent ideal conservative scenarios that deviate significantly from real atmospheric conditions, thereby leading to substantial discrepancies in comparison with the kinetic scheme. For instance, in the Pakistan event, the isentropic scheme yields more moisture originating from adjacent regions. This could be due to the initial $\theta$ underestimating the real $\theta$ during the air parcel's movement. Consequently, air parcels stay
in the lower troposphere where horizontal wind speeds are weaker, limiting their spatial displacement and confining them to a more localized region during successive time steps.

Based on the result of this test, we conclude that careful consideration must be given to the choice of the air parcels' vertical movement scheme. The two thermodynamic schemes (isentropic and equivalent potential temperature) are deemed unsuitable for moisture tracking, as they rely on conservative assumptions that fail to represent the complex thermodynamic
processes involved in moisture transport, particularly during precipitation. By contrast, the kinetic scheme has been shown to provide more physically realistic trajectories (Cloux et al., 2021), and it is therefore recommended. Accordingly, all subsequent tests will employ the kinetic scheme.

### 3.3 Parcel release height

As described in Section 2.4.3, the parcel release height represents the level at which the precipitation element originates.
Figure 4 illustrates the differences in moisture sources among the three release height options: panel (a) shows the difference between Option 2 (water vapour only) and Option 1 (all water species), while (b) shows the difference between Option 3 (cloud hydrometeors only) and Option 1. Corresponding results for the Pakistan and Scotland events are included in Fig. S7 and Fig. S8.





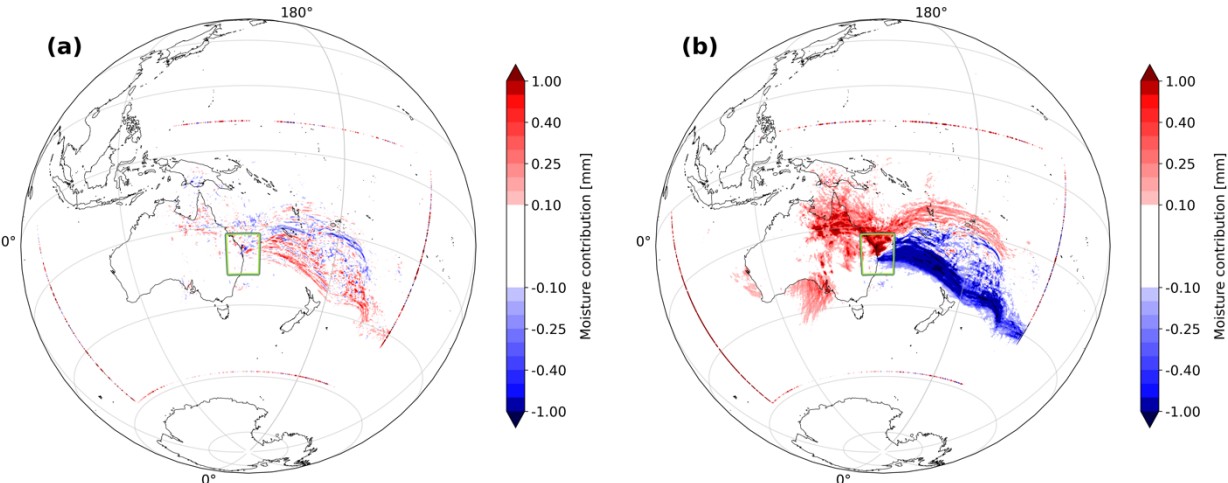

**Figure 4 Sensitivity of moisture source to different water species included in moisture profile in determining the parcel release**
**height. The difference of moisture sources between Option 2 (water vapor only) and Option 1 (all water species) for (a) and Option**
**3 (cloud hydrometeors only) and Option 1 for (b).**

Figure 4(a) shows minimal differences in moisture source results when the initial vertical position of air parcels is determined using water vapor profiles versus all water species profiles. In contrast, a substantial difference is observed when only cloud hydrometeors are used instead of all water species. A similar pattern is evident in the results for the Pakistan and

Scotland events (Fig. S7 and Fig. S8, respectively). These findings suggest that the initial release heights of air parcels strongly influence moisture source estimates, likely due to vertical wind shear between different atmospheric layers. For the Australia event, the majority of cloud water mass was concentrated between 600 hPa and 400 hPa. During the back tracking period, mid-level southwestly wind and eastly low-level wind (not shown here) resulted in significant differences in the horizontal movement of air parcels released at different heights. Consequently, more moisture is identified from the tropics

rather than midlatitude to the east when Option 3 rather than Option 1 is applied to determine the parcel release height. Tuinenburg & Staal (2020) also demonstrated that releasing air parcels based on vertical water vapor profiles resulted in larger latitudinal flows than releasing it from the surface. The test also highlights the predominant role of water vapor in the column total precipitable water, and underscores the distinct vertical distribution of cloud hydrometeors and water vapor.

During the target storm events, precipitation is generated through both the cloud microphysics parameterization and the

cumulus convection parameterization. Since ERA5 data are available at hourly intervals, but convective processes that lift and convert water vapour into cloud hydrometeors occur on shorter time scales, the "all water species" option provides the most comprehensive view of the atmospheric water that contributed to precipitation within each hour. Hence, Option 1 is recommended.



### 3.4 Well-mixed assumption and moisture identification method

In this test, two different identification methods are applied for along-trajectory moisture identification as described in section 2.4.4. While the trajectories remain identical, the assignment of moisture to points along the trajectory differs between the methods.

Figure 5(a) and Figure 5(b) present the moisture source identification results using the ET-mixing method and the WaterSip method, respectively, for the entire air column (i.e. with equivalent tracking within and above the PBL) during the Australia event. The ET-mixing method yields geographically diffuse moisture sources, whereas the WaterSip method highlights more localized moisture sources near the coast. A more pronounced difference is observed in the Pakistan event (Fig. S9), where the WaterSip method emphasizes specific humidity uptake in local regions, while attributing less moisture contribution to the distant southern Indian Ocean compared to the ET-mixing method. The Scotland event exhibits an even more striking contrast between the two methods. Notably, the WaterSip method clearly captures the structure of an atmospheric river, which has been identified as the primary mechanism driving this extreme event (Graham et al., 2023) . It is important to note that, in general, atmospheric rivers are not sustained by evapotranspiration directly beneath them, but rather by the convergence of moisture evaporated a significant distance away (Gimeno et al., 2016). We also note that WaterSip can generate moisture contributions that are higher than the evapotranspiration that occurred in a given location.

In fact, the specific humidity increase of an air parcel across a grid point results from both convergence and surface ET. For extreme events, the moisture contribution from convergence is expected to be much larger than that from surface ET (Wei et al., 2016; Zhao et al., 2020). Consequently, the WaterSip method will capture the convergence-related atmospheric river, but may not necessarily identify the original, surface evaporative sources of the atmospheric moisture. In contrast, the ET-mixing method records the percentage of moisture from below the current grid point relative to all moisture in the air parcel, and is thus more directly related to surface ET. This is supported by Crespo-Otero et al (2024), who applied both Utrack and WaterSip methods with back trajectories from WRF-FLEXPART to identify moisture sources in five atmospheric rivers. Using WRF-WVT output as the "ground truth", they found that Utrack gives more accurate results than WaterSip. This outcome is consistent with the fundamental difference between the two methods and aligns with our earlier reasoning particularly for events dominated by atmospheric rivers.

However, the "well-mixing" process should apply to all moisture within the air column, not merely to the ET from below the air parcel. This implies that, to accurately identify all moisture sources contributing to an air parcel, one must consider the origins of moisture throughout the full column in which the parcel currently resides. Such a requirement leads to a cascading, unresolved problem. That's also why Lagrangian models give less diffusive moisture sources than Eulerian methods (Li et al., 2024).The only approaches that can identify the true moisture sources are isotopic observations or online Eulerian moisture tracking methods, such as WRF-WVT— assuming that model parameterizations in WRF do not introduce significant errors. This limitation highlights a fundamental constraint of Lagrangian moisture tracking models. Nevertheless, with 'ET-mixing' identification method, Lagrangian methods can still provide valuable insights for research of surface




evaporative moisture sources, with the WaterSip method able to indicate the location of atmospheric moisture uptakes associated with studied precipitation events.

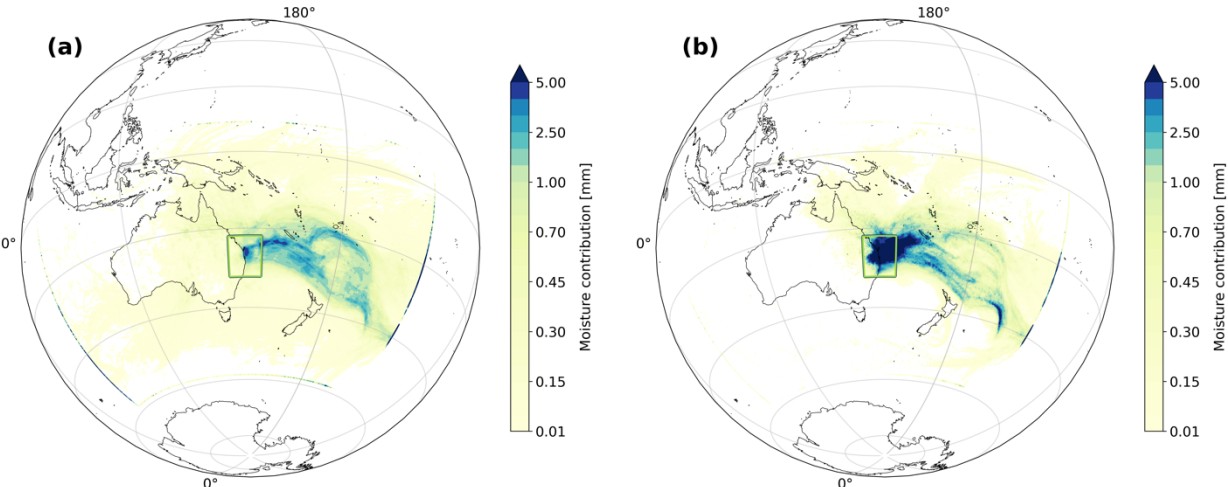

**Figure 5 Moisture Sources result of Using 'ET-mixing' method and "Water-Sip" methods for moisture tracking through the whole air column**

The applicability of ET-mixing method depends on the validity of the vertically well-mixed assumption. In some cases, surface ET is not well-mixed in the whole air column, but confined primarily to the lower atmosphere. As described in section 2.4.4, Option 3 addresses this by dividing the whole air column into two layers, within and above PBL. Figure 6(a-c),

Figure S11 (a-c) and Figure S12 (a-c) present the moisture source results based on Option 3 for the three selected events. A substantial fraction of moisture is attributed to convergence occurring above the PBL, where the true origin of moisture can not be identified through this approach. This is especially evident in the Scotland event in which an atmospheric river appears to play a dominant role.

When strong convection is present, the entire atmospheric column is often assumed to be well-mixed and the ET-mixing

method is applied throughout the whole column. This constitutes the distinction between Option 3 and Option 4. Figure 6(d-f), Figure S11 (d-f) and Figure S12 (d-f) illustrate the results differences between these two options. In both options, the trajectories of air parcels are identical, the differences in results arise solely from the choice of moisture source identification method under convective conditions when the air parcel is located above PBL. The influence of this difference to the result further propagates through subsequent steps. As demonstrated in the supplementary material, the discrepancies between the

two options are non-negligible, so convection is a factor that should be considered in moisture source identification as was earlier found in Forster et al.(2007). Based on these results and physical considerations, Option 4 is recommended.



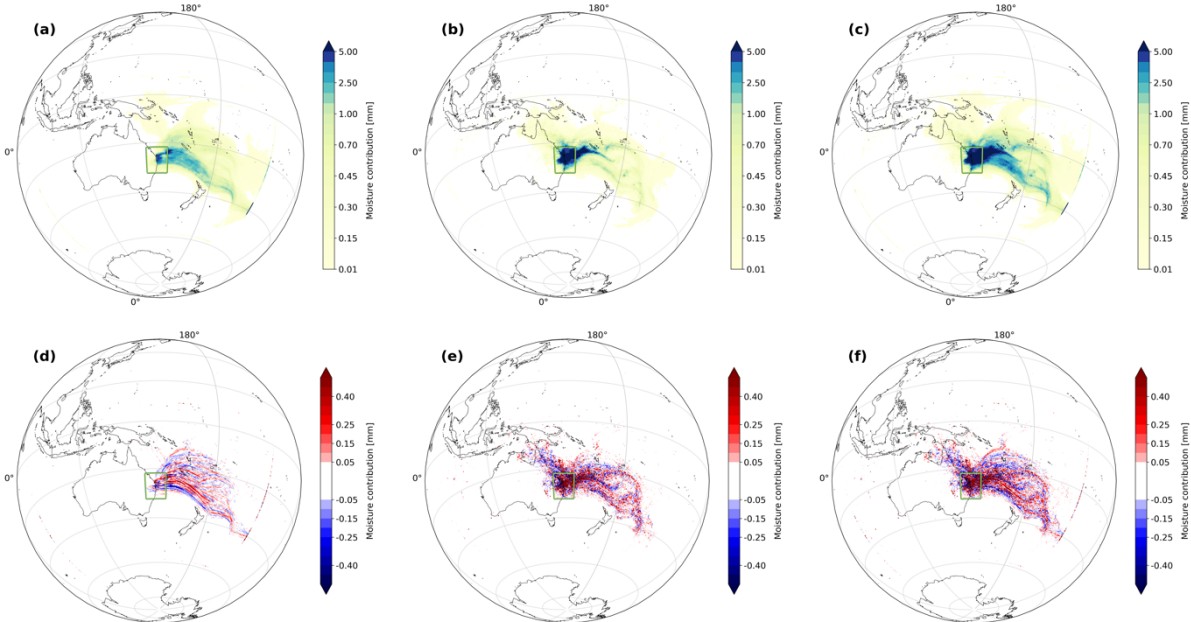

**Figure 6 Moisture source results of splitting the air column into (a) within PBL (uses 'ET-mixing' assuming well mixed) , (b) above PBL (uses WaterSip), and (c) the total, which is the sum of (a) and (b). The 2nd row shows the difference after adding consideration of convection, from left to right, moisture sources difference (d) when the atmosphere is well mixed ( uses 'ET-mixing' within-boundary layer and the whole atmosphere when convection occurs), (e) above PBL (uses WaterSip) and (f) the total, which is the sum of (d) and (f).**

## 3.5 Within-grid interpolation

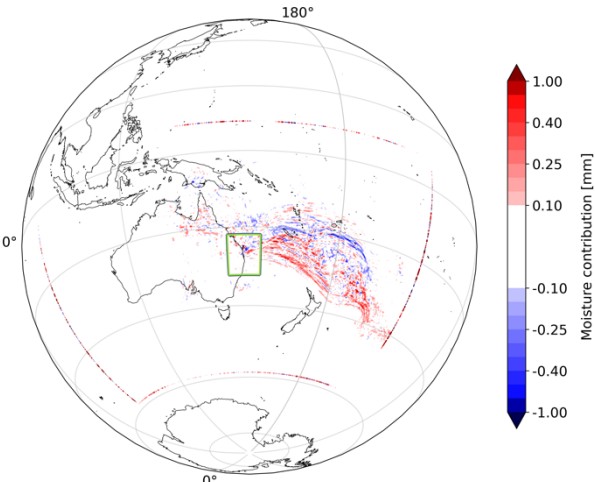

**Figure 7 Differences of moisture source results resulting from bicubic and bilinear interpolation**

The differences between the results caused by the choice of within-grid interpolation methods are minimal (Fig. 7; see Fig. S13 for the Pakistan and Scotland events) with a root mean square error of 0.10mm, pattern correlation of 0.982, for the





Australia event. Bilinear interpolation utilises 4 neighbouring points, whereas bicubic interpolation incorporates 16 points. The small discrepancies observed can be attributed to two main factors. First, the variable fields in the domains of the three

events exhibit relatively weak spatial gradient, so incorporating 16 points does not add substantially more information than 4 points. Second, the high spatial resolution of the input data ensures that both interpolation methods yield very similar results. This finding is consistent with the conclusion of Stohl et al. (2001) , which emphasize that the quality of the input data is more critical than the choice among similarly performing models. Tuinenburg & Staal (2020) also showed that the effect of linear interpolation of forcing data to moisture footprints is marginal compared with the 'nearest-neighbour value' method,

with 3% slower in computational speed. Given its computational efficiency (about 2 times faster than bicubic interpolation), bilinear interpolation method is applied in other sensitivity tests of this study and is also recommended for input data that with a high resolution.

**3.6 Back trajectory time step**

The temporal resolution of ERA5 reanalysis is 1 hour. In this study, four different back-trajectory time step options are

tested: 10, 15, 30 and 60 minutes.

The differences between the 10-minute and 15-minute time steps are negligible (Fig. 8a), while discrepancies become more apparent when comparing the 10-minute and 30-minute time steps (Fig. 8b). The deviation becomes particularly pronounced for the 60-minute time step, which exhibits substantial shifts in moisture sources, even after regridding (Fig. 8c). Specifically, the 60-minute time step causes some moisture sources to shift from north to south.

Dirmeyer and Brubaker(1999) used a 6-hourly resolution for wind and moisture inputs and concluded that a 1-hour time step was sufficient for numerical stability. Similarly, Crespo-Otero et al. (2024) opted for 3-hourly data despite the availability of hourly data, as extensively high temporal resolution was found to introduce noise into the WaterSip diagnostic, leading to systematic biases.

This study underscores the importance of selecting an appropriate time step when using ERA5 reanalysis data as input.

Based on the findings in this study and previous research, the optimal time step choice depends on both the temporal and spatial resolution of the input data, as well as the characteristics of the precipitation being tracked. Larger time steps can cause air parcels to skip multiple grid cells, particularly in regions with stronger winds, leading to inaccuracies in the representation of moisture sources. For ERA5 reanalysis, a time step in the range of 15 minutes is recommended.



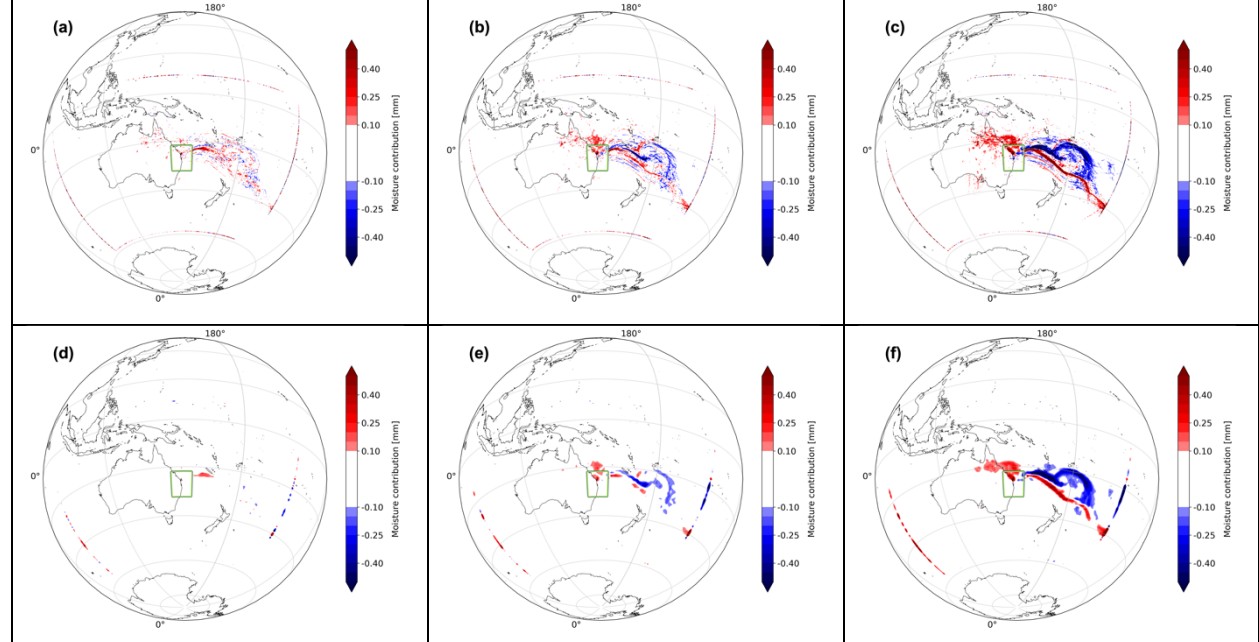

**Figure 8  Differences of moisture source results between time step 15min vs 10min(a), 30min vs 10min(b), 60min vs 10mins(c), regridded into 1 deg as (d-f) respectively.**

## 4 Conclusion

Lagrangian moisture tracking methods rely on assumptions that represent the thermodynamic and dynamic processes governing air parcels' movement. While these models have been widely applied in both individual events and climatological studies, their underlying assumptions have rarely been systematically tested or analyzed separately for larger applicability across different regions and precipitation types.  This study assesses these assumptions through sensitivity tests, aiming to identify the physical assumptions that have a greater impact on the moisture source tracking result, thus helping to improve Lagrangian models for moisture source identification.

The assumptions assessed in this study include: (1) the number of air parcels that need to be released for a study region, (2) the vertical movement schemes of air parcels, (3) the release height of parcels, (4) the vertically well-mixed assumptions and moisture identification method, (5) the within-grid interpolation method, and (6) the trajectory time-step.

The results suggest that releasing approximately 200 air parcels per day per grid point yields results with comparable accuracy (~0.95 correlation) to those obtained using 1000 parcels,  offering a more computationally efficient setup without compromising reliability. The choice of vertical movement scheme of air parcels is critical: the kinetic scheme is recommended, while quasi-isentropic flow and flow along equivalent potential temperature represent two thermodynamic extreme approximations. These two schemes are not recommended due to their event-dependence and potential deviation from real atmospheric behaviour. Parcel release height also significantly affects the results. All atmospheric water species



(both water vapor and cloud water species) should be considered in precipitation formation for the humidity profile, thus to determine the parcel release height, because different types of moisture can be transformed into each other. The vertically well-mixed assumption is tested through two along-trajectory identification methods, namely 'ET-mixing' and 'WaterSip'.

Results indicate that the WaterSip method tends to favor local sources and underestimate remote moisture contributions, and instead of directly estimating surface evaporative moisture sources, it reflects regions of strong air convergence, especially during atmospheric river-related precipitation events. In comparison, the ET-mixing method more directly identifies regions where moisture originates via evapotranspiration from land or ocean surfaces, providing insights about how precipitation might respond to changes in surface ET or atmospheric circulation. Considering well-mixing is theoretically thought to be

within PBL rather than the whole air column, ET-mixing and WaterSip are applied within and above boundary layer respectively, finding that a quite large percent of moisture for studied precipitation coming from convergence above PBL. Furthermore, if deep convection is thought to make the whole column well-mixed (well-mixing method is applied even above PBL), the result of moisture sources also changes. Finally, for the moisture identification, air column dividing and convection are both considered. The within-grid interpolation method appears to be less important in this study, based on the

use of ERA5 reanalysis as input, as the spatial resolution of ERA5 reanalysis is sufficiently high. Lastly, the time step is proved to be influential when exceeding a certain threshold. For ERA5-driven Lagrangian calculations, a time step of 15 minutes is recommended to maintain precision.

This study systematically tests the sensitivity of key assumptions for identifying precipitation moisture sources using Lagrangian moisture tracking methods, offering a reference for the application of backward Lagrangian approach in moisture

source identification. Further studies are encouraged to further explore and constrain the most influential assumptions identified in this study.

*Code availability*

The Lagrangian model BTrIMS used for this paper is available at https://github.com/YinglinMu/BTrIMSv1.1_original/ and

is archived on Zenodo (https://doi.org/10.5281/zenodo.15703767 ,Mu, 2025).

All data used in this study are publicly available. ERA5 reanalysis hourly data (Hersbach et al., 2020) for pressure levels (https://doi.org/10.24381/cds.bd0915c6, Hersbach et al., (2023a) ) and single levels (https://doi.org/10.24381/cds.adbb2d47, Hersbach et al., (2023b) )  can be downloaded from the Copernicus Climate Data Store.

*Author contributions.* YM, JE and AT conceived the study, YM, JE and CH created the Fortran framework available on Github. JE and YM designed the experiment. YM ran the simulation and performed the analysis. YM designed the layout of the paper and leading the writing. All authors have been involved in interpreting the results, discussing the findings and editing the paper.

Competing interests. The contact author has declared that neither she nor the coauthors have any competing interests.



*Acknowledgement*

This research was supported by the Australian Research Council Centre of Excellence for the Weather of the 21st Century
(CE230100012) and was undertaken with the assistance of resources from the National Computational Infrastructure (NCI
Australia), an NCRIS enabled capability supported by the Australian Government.

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
