# Peer review of "Enhancing the Lagrangian approach for moisture source identification through sensitivity testing of assumptions using BTrIMS1.1"

_EGUsphere, 2025_

## Author Comment (AC1)

**Reply to comments from referees**

**RC1**:**

This study investigates the sensitivity of the Lagrangian estimation of moisture sources for precipitation to the choice of different assumptions and configurations, using the model BTrIMS. The authors explore several factors that could guide researchers in selecting appropriate settings for moisture tracking, such as the number of parcels released, the time step in the trajectory method, the initial vertical distribution of parcels, and the influence of different interpolation methods or mixing schemes. Although only one model is used, the results of some of the experiments are easily extrapolated to other models. Overall, the study is well-defined, clearly presented, and well written. The main area of improvement lies in the presentation of results, as in the absence of a general ground truth it is difficult to identify the optimal configuration from certain experiments. Furthermore, presenting the results for the three analyzed cases, together with a more rigorous exposition of the methodology, would help readers follow the conclusions more easily, as elaborated in the general and specific comments below.

**General comments**

1. Lack of a reference configuration for comparison. Except for the test involving the number of air parcels released per grid point, there is not a reference configuration against which to compare the results. It would be beneficial to define a standard or "reference configuration" of BrTrIMS, which would be the most complex and realistic one, and then analyze how relaxing specific assumptions or modifying different components of the configuration impacts the results. For instance, the reference configuration could involve releasing 1000 air parcels per grid point, using the wind field for the vertical movement (kinetical scheme), using the ET-mixing method for moisture tracking, applying bicubic interpolation, and employing the smallest time step. Using this approach and clearly stating the reference setup in Sect. 2 would help the reader in following the analysis, as I find it difficult to determine the exact configuration that is being used in each subsection of Sect. 3. For example, what is the time step in Sect. 3.1? How many parcels per grid point are released in Sect. 3.6? Additionally, in cases where it is unclear which configuration is more realistic (such as choosing between the ETmixing approach and WaterSip for moisture tracking) I would avoid making categorical statements about one setup underestimating results compared to the other, since there is not a "ground truth" for direct comparison in these cases.

**Reply:** For tests in which the configuration was not clearly described, we will detail the configuration.

Since there is no "ground truth" to validate the results, terms like "underestimate" or "overestimate" are not correct. Thanks for pointing this out. Instead we will note where one method is lower than another.

2. Focus on the Australian event. Although this study analyzes three different precipitation events (Australia, Pakistan and Scotland cases), the main manuscript presents results only for the Australian event. I believe that in most subsections it would be feasible and beneficial to include results for all three events, either by presenting them side by side or by averaging. For instance, in Sect. 3.1, the results shown in Fig. 1 could be averaged across the three events, which would provide the same information. In the case of Fig. 2 averaging may provide less valuable information, but it would still be possible to show the curves for all three events using a single value of pattern correlation, and move the analysis for different values of the pattern correlation to the supplement. Similarly, the subsequent maps could be easily displayed for all three events together, as there is minimal overlap between them.

**Reply**: Thanks for this comment. We believe that averaging the three cases would result in a loss of important information. Therefore, we decided to present the results for the three cases independently rather than as an average. However, we will add result of the other two cases in Figure 1. For example, for Figure S1a first panel, we can take the average pattern correlation per number of grid points. This will give a line with one correlation (y-value) per number of grid points (x-value) and we can also add the line of Pakistan and Scotland cases.

The main manuscript will be too long and harder to follow if we put all figures of the three cases into it, so we will select the most critical figures for the Pakistan and Scotland cases that contribute to the main manuscript results and conclusions, while avoiding increasing too much the article's length and complexity. For example, we will include Figures S5, S6, S7, S8, S9, S10, while keeping the other figures for the Pakistan and Scotland cases in the Supplement.

**Specific comments**

L59-1: COSMO is a meteorological model used for numerical weather prediction and atmospheric research, not an Eulerian method for moisture tracking. In Winschall et al., (2014) they use an Eulerian tagging approach implemented in this model. It would be useful to clarify this information.

**Reply:** Thanks for noting. COSMO itself is a numerical prediction model, we have corrected this accordingly.

L59-2: The water vapor tracers implemented in the WRF model are most commonly abbreviated as WRF-WVTs. Also, the correct article to cite is Insua-Costa et al., (2018), where this tool is presented and validated in detail.

**Reply:** In both papers, the water vapor tracers implemented in the WRF model is abbreviated as WRF-WVT rather than WRF-WVTs. Therefore, we prefer to keep it as WRF-WVT here for consistency with those papers.

Yes, in the paper by Insua-Costa et al. (2018), the formulation and implementation details of the method are provided. We will correct the reference to this article, thank you for the suggestion.

L60-61: Here it is asserted that Eulerian moisture tracking methods "are precise". While this is true in general, the accuracy of these methods depends on how well the meteorological models in which they are implemented represent reality. It is possible to have a model simulation very deviated from reality, and then the moisture source calculation would be accurate in the model world, but not in reality. In this case, a moisture tracking method using reanalysis would be more accurate. Please, clarify that water vapor tagging methods depend on the accuracy of the underlying meteorological model.

**Reply:** Thank you for pointing this out. Yes, it is not entirely correct to say that Eulerian moisture tracking methods are precise. We have added a clarification that such methods are precise only when the meteorological models itself can accurately represent reality.

L75: For accuracy and rigor, please use this more-explicit form of the trajectory equation d**X**/dt=**u**[**X**(t)], where the full dependence on time is highlighted. The velocity field **u** may also depend on time, not only on the 3-dimensional position. Furthermore, although it is useful to interpret dx and dt as the air parcel's displacement in one time step and the time step, from a mathematical point of view it is not correct to state that, since d**X**/dt is just the derivative of **X** with respect to time.

**Reply:** we thank the reviewer for this clarification. We agree that, from a strict mathematical perspective, dX/dt is the derivative of position with respect to time, and dx and dt are not finite quantities representing displacement and time interval. In the revised text, we have clarified that dX/dt = u(X,t) represents the air parcel's velocity, and in the numerical implementation the displacement over one time step is approximated as  $\Delta X = u(X,t) \Delta t$ .

L79-81: This may lead to misunderstanding, as it appears that Dirmeyer and Brubaker, (1999) do not use the wind fields at all. Please rephrase to indicate that wind fields are used in trajectory calculations to drive the horizontal movement of the parcel.

**Reply:** We thank the reviewer for pointing this out. We will clarify in the revised text that Dirmeyer and Brubaker (1999) only modified the vertical movement scheme of air parcels, whereas the horizontal advection is still driven by the wind fields.

L84: FLEXPART also includes a detailed description of turbulence.

**Reply:** Yes, FLEXPART also considers turbulence, and we have added FLEXPART here.

L97: Typo in "These two identification methods also differS in this".

**Reply:** Thanks, we will change "differs" to "differ".

L100: "Due to limitations in Eulerian methods for moisture tracking". It would be useful to expand on these limitations. In L61 the need to predefine water source regions is mentioned. What about the computational requirements of these methods?

**Reply:** We will expand this section to provide explanations of the limitations of both offline and online Eulerian methods, including the need to predefine water source regions and the high computational requirements, which make these methods less practical in certain applications.

L113: "there are three schemes based on different theories". Please, refer to the section/subsection where these schemes are introduced and explained.

**Reply:** Yes, we will refer section 2.4.2 where the three schemes are introduced.

L136-137: "The air parcels are advected by wind" suggests a forward tracking of air parcels. Please rephrase for clarity. Also, here and in other parts of the manuscript, there is a reference to a "predefined large domain", but it is not stated anywhere what this domain is (it may be deduced from Fig. 4). It would be useful to include a visualization of the domains in the appendix.

**Reply:** Thanks for these points, the relevant descriptions will be rephrased for clarity. Regarding the "predefined large domain", it is user-defined by setting the boundaries in latitude/longitude; we will clarify this in the text and highlight it in a figure to give readers a more intuitive understanding.

L173-L189: Although the description of the Australia event is very complete and detailed, I would move it to the appendix, as it may distract the reader from the main focus of the paper. I would only include a small summary with the most essential characteristics of the event, and perhaps also include a small summary of the other two cases.

**Reply:** Thanks for this comment. We think it is necessary to let readers know about the synoptic features of the three cases, so we want to keep a basic description of the three events in the main manuscript. We will bring text S1 and S2 about the introduction of Pakistan and Scotland cases into the main manuscript for completeness and context for the events, and we will shorten the synoptic description of the three cases to avoid distracting the reader from the main focus. Review2 suggests adding synoptic charts for the three cases, we will accept this suggestion, but will put the charts into supplementary material.

L200-201: "total-precipitable-water-weighted height". The total precipitable water is a two-dimensional field, calculated as the integral of all water components in the atmospheric column. Thus, this expression may lead to misunderstanding. If parcels are released randomly vertically following the humidity profile, please replace "total-precipitable-water" with "humidity", otherwise explain how parcels are released in more detail.

**Reply:** Thank you for pointing this out. We agree that the term "total-precipitable-water-weighted height" is misleading. We will revise the text accordingly and replace it with "humidity-weighted height" to clarify the method of parcel release.

L225-L238: I do not see the point of having this equation and all the involved parameters here. I would consider moving it to the appendix.

**Reply:** Thanks for your feedback. We included this equation here because there are other equations for calculating the equivalent potential temperature based on different theories. Although the equations are similar, there are small differences among them. However, we agree that this equation could be moved to the supplementary material to improve the flow of the main text, and we will do it in the revised manuscript.

L276-278: Option 2 does not impose a threshold on initial relative humidity or require moisture uptake to occur within the PBL, arguing that subgrid processes can allow the lower troposphere to contribute to precipitation. This is true in general, but it also depends on the chosen time step. Since the time steps used in this study are short (less than 1 hour), I believe that the initial relative humidity threshold may have an important effect on the results. It would be useful to include these results in the appendix, even if the impact on the results is less important than expected.

**Reply:** Thank you for the detailed feedback. Since imposing a threshold on the initial relative humidity and requiring moisture to occur within the PBL is the "standard" configuration of "WaterSip", we agree that it would be useful to present the results of this configuration, and we will include them in the Supplementary material.

L291-305: The only difference between the first set of equations and the second is in  $frac_{n1}'$  and  $frac_{n2}'$ , as  $fac_q$  may be updated as  $fac_q$ (1-  $frac_{n1}'$ ). Considering this can help

reduce the number of equations here and also to explain the differences between both sets of equations. If left as is, I would move the equations to the appendix and explain them in detail there.

**Reply:** Thank you, yes, the only difference is between fracn1' and fracn2'. We will make the suggested simplification, but leave it in the main manuscript because they are important for understanding the two methods: "ET-mixing" and "WaterSip".

L318: It would be useful to clarify here if parcels are also released from every grid point where precipitation occurs every time step, or if they are released with less frequency (for example, every 1 or 3 hours).

**Reply:** Great point. In this study, we perform temporal interpolation of precipitation to the back-trajectory time step. This means that the temporal profile of precipitation, which determines the parcel release time, is adjusted accordingly. Parcels are released at every back-trajectory time step where precipitation occurs (above a minimum threshold). We will revise this in the new version.

L321-322: I understand that pattern correlations are calculated by computing the Pearson correlation coefficient between two spatial distributions of moisture sources: the "true" one (1000 parcels) and the tested one (50, 100, 200 and 500 parcels). If this is the case, please explain it in more detail. Otherwise, explain how pattern correlation is calculated.

**Reply:** Yes, your interpretation is correct. This will be clarified in the text.

Furthermore, I think it could be better explained how the results in Fig. 1 are obtained. If I understand correctly, the Australian event involves a certain number of points (around 1000) where precipitation is larger than 1 mm, and then, for each given number of grid points, a smaller region of this size is being selected to calculate the pattern correlation. The smaller the number of grid points, the greater the number of regions that can be selected, and therefore the variability in pattern correlation decreases with the number of grid points.

**Reply:** Yes, your understanding is correct. We will explain this process more clearly in the revised version.

L351: "Since we only selected four air parcels per grid point". I understand that you are referring to the maximum number of air parcels per grid point (50, 100, 200 and 500). Thus, it would be more accurate to say "four different numbers of air parcels per grid point".

**Reply:** We have corrected this as suggested.

L487: Shouldn't it be "(e)" instead of "(f)"?

**Reply:** A typo, it has been corrected.

L540: Due to the absence of a reference for comparison, I would not use words such as "underestimate". It is true that WaterSip gives more importance to local sources than other methods, but here there is not a reference methodology with which to compare the results, so it cannot be said whether the correct results are those of WaterSip or those of BrTrIMS.

**Reply:** Yes, we will remove this word "underestimate", as we don't have a "ground truth" to validate it. We will instead note where one result is lower or higher than the other.

L546: Shouldn't it be "comes" instead of "coming"?

**Reply:** Yes, we have rephrased this sentence.

L548: What is the meaning here of "air column dividing"?

**Reply:** "air column dividing" means that "ET-mixing" is applied when the air parcel is within boundary layer, and the moisture source is identified directly from the evaporative sources, above boundary layer, the "WaterSip" method is applied, moisture source obtained from this is regarded as above PBL moisture additions, not evaporative moisture sources. When there is a deep convection, "ET-mixing" is applied either within or above PBL. We will make the summary clear here.

**RC2**

This study uses a variety of sensitivity experiments to assess the uncertainty of moisture source calculations based on methodological assumptions. These experiments cover different setups for trajectory calculations, dataset resolutions and moisture source diagnostics for three case studies. For all experiments, the authors provide recommendations for choosing parameters and diagnostics. The study is well-written and logically structured.

While this study investigates important questions regarding the uncertainty of Lagrangian moisture source identification, it focuses mostly on one moisture source diagnostic (ET-mixing) and relies on previous studies for many recommendations. While the authors imply in the introduction that the new aspects of this study are the combination of several Lagrangian moisture source diagnostics with several case studies and precipitation types, the results focus mostly on an Australian case study and the ET-mixing method. The study could be improved by integrating the three case studies better, extending the sensitivity experiment to test more assumptions for other methods than ET-mixing and better justifying the recommendations and choice of setup.

**Main comments:**

1. Case study introduction and discussion: In section 2.3, three case studies are introduced, highlighting that "an assumption that is physically meaningful in one region or specific type of precipitation events may deviate significantly from reality in another region or under different meteorological conditions." This is an important aspect to investigate, but only the Australian case study is introduced in the paper with some detail, while the other two remain in the supplement. Further, no synoptic charts of the case studies are shown. The comparison of these case studies takes little space in the results and discussion sections, even though it is implied in the introduction that the comparison of events from different climatological regions is a new aspect of this study compared to previous studies (see lines 100-104). To understand the differences between the moisture source diagnostic, it'll help to have a better description of the synoptic processes, instead of climate modes, and e.g. vertical wind shear, vertical moisture structure and boundary layer height.

**Reply:** Thank you for the comment. We agree that giving a short summary of Pakistan and Scotland cases in the main manuscript is a good idea, together with synoptic charts, which we will place in the supplementary material to avoid increasing the manuscript's length too much.

In terms of figures for Pakistan and Scotland cases, we do think that moving some figures from the supplement to the main manuscript would help in the

comparison of the three cases, for example, Figures S5, S6, S7,S8,S9, S10. However, we we will keep other figures in the supplementary material, as these figures shows similar results to the Australia case.

For comparison among the three cases, we do have relevant content in both results and discussion. For example, lines 345-350 discusses the different number of air parcels required for different vertical movement schemes in the three cases; lines 377-388 discuss the influence of vertical movement scheme is different for different cases.

2. Comparison of moisture source diagnostics: Large parts of the sensitivity experiments are based on a moisture source diagnostic based on the ET-mixing assumption. For one sensitivity experiment, the WaterSip method is used in the BTrIMS1.1 setting, but no sensitivity tests are done on the assumptions going into the WaterSip calculations (e.g. RH threshold or deltaq threshold). Further, the parcel release height is based on a different method (vertical humidity profile) than commonly used in WaterSip (a combination of detlaq and an RH threshold). In this current form, I would not call this study a comparison of several Lagrangian moisture source identifications, as it mostly focuses on one method and does not use standard setups for other methods. Therefore, results for WaterSip from this study are not comparable to many other studies.

**Reply:** Yes – most tests have been done using the ET-mixing assumption. We will remove any reference to "several" methods and refer instead to "alternate moisture source identification methods" when appropriate. We will also test the standard "WaterSip" setting to enable comparison of other studies.

3. **Physical concepts:** Physical processes and concepts, and decisions based on them, could be explained better, e.g. why an RH threshold to detect the cloud layer (layer of origin of precipitation) is worse than using a vertical water vapour profile (section 2.4.3), "well-developed cyclonic boundary" (lines 372-373), moisture convergence and its effect on Lagrangian tracking of moisture, WaterSip and the well-mixed method (line 445ff), the effect of combining ET-mixing and WaterSip on the water budget along the trajectory (Option 3 and 4 in section 2.4.4).

**Reply:** Thank you for your comment, we will add further explanation to the text. The vertical precipitable water profile is preferred to the RH threshold (section 2.4.3) as it accounts for vertical variations in the total moisture available for precipitation formation. This is explained in more detail later in this reply, but we will also make it clear in section 2.4.3.

I used "well-developed cyclonic boundary" to show that choosing kinetic scheme for vertical movement is realistic, because it can show the synoptic

processes in the Australian case. However, without detailed introduction of synoptic conditions, it's abstract to say this, we will remove this sentence: "Another difference lies in the presence of a well-developed cyclonic boundary within the main moisture source regions under the kinetic scheme, both to the north and south. This boundary becomes less distinct under the isentropic scheme and appears even more diffuse under equivalent potential temperature scheme. "

For line 445, we will also clearify that the WaterSip method detects both bottom boundary ET, but also side boundary convergence, and this side boundary convergence contribution can be higher than ET.

4. Recommendations: Except for the recommendation on the number of parcels released per day, the recommendations on the best setup of the diagnostics should be better motivated. Without knowing the true moisture sources (where it is not known if the ET-mixing or WaterSip method is closer to the truth), the recommendations are often based on theoretical concepts or previous studies, while it is not clear how the sensitivity tests inform these recommendations.

**Reply:** Thanks for your comment. In the absence of an observed truth, many recommendations cannot be stated categorically. Here we evaluate these sensitivity tests, theoretical understanding, and past studies to inform our recommendations.

In the discuss section, we have given theoretical basis of these assumptions and recommendations.

For vertical movement scheme: line 398- "The two thermodynamic schemes (isentropic and equivalent potential temperature) are deemed unsuitable for moisture tracking, as they rely on conservative assumptions that fail to represent the complex thermodynamic processes involved in moisture transport, particularly during precipitation."

For parcel release height: line425- "Since ERA5 data are available at hourly intervals, but convective processes that lift and convert water vapour into cloud hydrometeors occur on shorter time scales, the "all water species" option provides the most comprehensive view of the atmospheric water that contributed to precipitation within each hour. Hence, Option 1 is recommended".

For "ET-mixing" and "WaterSip" method, we discussed the process they represent, from line 445-464.

We will further clarify how the sensitivity tests have contributed to recommendation in each case in the revised version.

**Detailed comments:**

**Title:** What do mean with "enhancing" the Lagrangian approach?

**Reply:** In this case, "enhancing" means analyzing the Lagrangian approach thoroughly, testing the assumptions systematically, and making some concepts clearer to support moisture tracking research.

We think that it would be better to change "Enhancing" to "refining".

**Line 22:** 200 air parcels per day -> at different heights? or different time steps? Per grid point?

**Reply:** 200 air parcels is the total number of air parcels per day per grid cell from different heights. We added a sentence after this one that the release time is based on precipitation rate while how to determine the release height is clearly explained later.

The original sentence is "We find that releasing approximately 200 air parcels per day from each grid point, is necessary to obtain accurate results for a region of 10 grid points or more (an area of ~9,000km2 in this case). "

We added this after it: "The distribution among different time steps in a day is determined based on precipitation rates."

**Line 26:** "the mechanisms behind these assumption" -> do you mean the mechanism that lead to different moisture sources based on different assumptions in the moisture source diagnostic? Consider rephrasing.

**Reply:** We mean "theoretical basis" by "the mechanisms behind these assumption", We will change "the mechanisms behind these assumption" to "the theoretical basis of these assumptions".

**Line 26**: "heat exchange" between air parcels? Or between the surface and the atmosphere?

**Reply:** Here, "heat exchange" refers to heat exchange between a specific air parcel and the surrounding environment.

However, we did not consider heat exchange processes in this study, so it may be clearer to remove these words to avoid misunderstanding.

The original sentence is: "The theoretical basis of these assumptions involve heat exchange, precipitation formation height, vertical mixing of surface evapotranspiration, and numerical noise, all of which must be carefully considered for realistic results.", we will remove "heat exchange".

**Lines 57-58:** "The tracking process is incorporated into the model in parallel with the water accounting process." What do you mean with "water accounting process"?

**Reply**: By "water accounting process", we mean water budget of air parcels along the trajectories, that is, assigning the moisture in an air parcel to the points along the trajectories.

We will change it to "parcel water accounting process" for clarity.

**Lines 59-60:** "The former is primarily used for climatological studies, while the latter is more frequently employed for regional research requiring higher resolution." Can you provide more references for this statement? COSMO tag is also used for regional high resolution simulations (e.g. the cited study by Winschall et al. (2014) is a regional case study).

**Reply:** Thanks for point this out. Yes, the original statement is not correct, COSMO tag is also used for regional high resolution simulations, so this sentence will be deleted.

**Line 74:** Equation 1 is not yet well described and integrated in text.

**Reply:** Reviewer1 also raised the same question about this, we have replied and will revise accordingly in future version of manuscript.

Origical text:

$$\frac{Dx}{Dt} = u(x) (1)$$

Dx: air parcel's displacement in one time step;

*Dt*: time step;

u(x): velocity field

It's changed to:

$$\frac{dX}{dt} = \mathbf{u}(x(t), t) (1)$$

Here, u(x(t),t) represents the velocity of the air parcel. In the numerical implementation, the air parcel's displacement over one time step ( $\Delta X$ ) is approximated as  $\Delta X = u(X,t) \Delta t$ .

**Line 84: Also FLEXPART considers turbulence.**

**Reply:** Yes thank you, we added FLEXPART in this sentence.

**Line 93**: "as moisture sources" -> as surface moisture sources?**

**Reply:** In Line 34, we define "moisture sources" as the evaporative sources from land or ocean surfaces. To make this clearer, we will change to "surface moisture sources".

**Line 94:** Sodemann (2008) -> Sodemann at el. (2008). This reference has been wrongly formatted in several places.

**Reply:** Thank you for pointing this out; the citation has been corrected.

**Line 119:** Can you provide a reference for cubic interpolation in Lagrangian studies?**

**Reply:** In line 308, we provided a reference, Stohl et al. (2001), the complete reference is: Stohl, A., Haimberger, L., Scheele, M. P., and Wernli, H.: An intercomparison of results from three trajectory models, Meteorological Applications, 8, 127–135, https://doi.org/10.1017/S1350482701002018, 2001. In this study (Stohl et al. (2001)), three Lagrangian models are compared, and the horizontal interpolation method of the three models are different, FLEXTRA used bicubic interpolation horizontally.

**Line 123:** What do you mean with subprocesses?**

**Reply:** There are two major subprocesses for Lagrangian approach in moisture source identification: 1) calculation of trajectories; 2) moisture identification along trajectories of air parcels.

We clarify it at line 66:" To clarify, there are two important sub-processes of Lagrangian methods for moisture tracking: calculation of trajectories and identification of moisture sources along trajectories."

**Lines 155-156:** "Temperature (T) is used to calculate potential temperature ( $\theta$ ) and equivalent potential temperature ( $\theta$ e)." -> also water vapour mixing ratio and pressure are needed to calculate the equivalent potential temperature.

**Reply:** Yes, we change this sentence: "Temperature (T) is used to calculate potential temperature ( $\theta$ ) and equivalent potential temperature ( $\theta_e$ )." to: "For calculation of potential temperature ( $\theta$ ) and equivalent potential temperature ( $\theta_e$ ), temperature (T), water vapour mixing ratio (q) and pressure are used.

**Line 168:** Can you be more specific what you mean by different types of precipitation events?

**Reply:** By "different types of precipitation events", we mean that the weather systems that cause precipitation events are distinct. For example, some are caused by a low-pressure system, while the others are caused by an atmospheric river. This will be clarified in the text.

At the end of section 2.4, we will add this sentence: "These three precipitation events are influenced by different weather systems, and thus they provide an opportunity to assess the robustness of these assumptions across a broader range of conditions."

**Section 2.3.1:** Climate modes are important drivers of interannual variability. To understand differences in the moisture sources, a synoptic-scale description of the events would be more helpful.

**Reply:** Yes, we will give more detailed synoptic-scale description as well as charts, but the charts will be put into supplementary material to avoid increasing too much the article's length and complexity.

**Section 2.3.2**: A description of how they differ with respect to precipitation type and synoptic settings would help to better understand differences between the events.

**Reply:** Thanks for this comment. We agree that giving description of synoptic settings will help to better understand differences between the events. We will give a basic introduction of the three cases, with synoptic charts put in the supplementary material.

**Lines 215-216:** "Within the boundary layer, potential temperature is modified by convective diabatic processes." This sentence seems disconnected from the method description. Can you clarify, why this is important for quasi-isentropic movement?

**Reply:** Through this sentence, we are just saying potential temperature can change due to diabatic processes. However, it should not be put in the method description, we will delete this sentence in the manuscript.

**Line 225:** Equation 2: Consider removing these equations as they are commonly known.

**Reply:** As there are different equations about calculating the equivalent potential temperature, and I use that by from Emanuel (1994), so I think it would be better to show it in the study. However, we will move this to supplementary material.

**Section 2.4.3:** Why is a vertical profile of precipitable water a good approximation of precipitation formation heights? The specific humidity is mostly highest in the boundary layer, close to the surface, but precipitation forms at elevated heights

upon lifting. Tracking precipitation based on the vertical humidity profile might overrepresent boundary layer moisture that does not contribute to precipitation.

**Reply:** The fundamental reason is that the boundary layer (or the whole air column if there is deep convection) can be well-mixed. Given that the time scale of the mixing process is smaller than a time step, moisture within the boundary layer can be convectively mixed into clouds within the atmospheric model output time step (hourly here) and contribute to precipitation.

**Line 258:** Can you elaborate on limitations of the well-mixed assumption?**

**Reply**: The limitations of well-mixed assumptions are discussed in paragraph 4 of section 3.4. The main limitation is the assumption that the entire atmospheric column is well-mixed is only true within deep convective cells.

**Line 291**: Many variables in these equations are not introduced.**

**Reply:** Noted, we will introduce all variables in these equations. Review 1 also gave suggestion about simplifying it and we accepted it.

**Figure 1: The term "kinetic" has not been introduced in the methods section**

**Reply:** We added this explanation in section 2.4.2, letting readers know that 'kinetic' means using vertical wind field.

line 212, "However, for vertical motion, there are primarily two different assumptions. The first assumes that vertical motion, like horizontal advection, is forced by the vertical wind field.", will be changed to "However, for vertical motion, there are primarily two different assumptions. The first assumes that vertical motion, like horizontal advection, is forced by the vertical wind field, which is called a kinetic scheme in this study.

**Section 3.1:** Can you introduce better how the number of parcels relate to the number of grid points?**

**Reply:** We will clarify the description of the test in section 2.4.1 and revise, the discussion of the results in here, making it clear that how Figure 1 is obtained and the conclusion from Figure 1.

**Lines 332-333:** "However, a minimum threshold for pattern correlation of each individual grid point must also be considered." Why "must"?

**Reply:** Individual grid point is the smallest unit for which moisture source identification is performed. Although a study region often includes tens or hundreds of grid points, when conducting city-scale research, accuracy of a single grid point

should ideally be considered. We think it would be better to change "must" to "should".

**Lines 349-351**: "Note that the curves in Fig. 2, Fig. S2, and Fig. S4 are intended as qualitative illustrations only, aimed at showing the relative differences among the various vertical movement schemes of air parcels, since we only tested four air parcels per grid point." What do you mean by qualitative illustrations? Are these not quantitative results from the sensitivity experiments?

**Reply:** Thanks for your comment. You are right, this statement is not clear. Actually in our study, the result is a quantitative one. We sought to make clear that the conclusion "200 air parcels per grid per day" is enough for regions with tens to hundreds of grid cells for most cases. However, for some regions or some cases, it can be slightly different.

**Section 3.2, 3.5:** What is the setup of the diagnostic (apart from the vertical movement)?

**Reply:** We will add clarification for the whole set up of each sensitivity test.

In tests 3.2-3.6, 200 air parcels are released per grid per day.

In tests 3.3-3.6, vertical wind field is used for air parcel's vertical movement.

In all tests except for 3.3, parcel release height is determined based on the vertical profile of all water mass, including water vapor and cloud hydrometeors.

In all tests except for 3.4, "ET-mixing" method is used for moisture identification.

In all tests except for 3.5, bilinear interpolation method is used.

In all tests except for 3.6, time step is 15 minutes.

**Line 394:** "This could be due to the initial  $\theta$  underestimating the real  $\theta$  during the air parcel's movement." What is the real  $\theta$ ? From observations and not from reanalysis?

**Reply:** By "real" I mean that  $\theta$  that the air parcel would have in the model world if the model had a tagging function.

We will change this sentence to "This could be due to an initially low estimation of theta without considering t diabatic processes."

**Lines 425-428:** ERA5 runs at a smaller time step than 1-hourly output and parameterises convective processes. Thus, it's true that it does not fully resolve convective processes. But in the hourly output, the instantaneous cloud properties

are provided, which represent the cloud location in the model world. As all calculations are based on the reanalysis data, relying on model output for the vertical extent of the cloud, seems reasonable. Therefore, I don't understand why Option 1 (including also water vapour) better represents the location of precipitation formation than Option 3.

**Reply:** Thanks for this comment. We are calculating the vertical distribution of parcels to release that produced the precipitation that fell throughout the hour – not just a snapshot of rainfall at the end of the hour. Thus, these instantaneous cloud properties are indicative only. It does not account for variations in the vertical profile within the hour or for moisture being convected from the boundary layer into the cloud base over that hour.

Lines 447-450: "Consequently, the WaterSip method will capture the convergencerelated atmospheric river, but may not necessarily identify the original, surface evaporative sources of the atmospheric moisture. In contrast, the ET-mixing method records the percentage of moisture from below the current grid point relative to all moisture in the air parcel, and is thus more directly related to surface ET. " Can you explain what you mean by convergence? Large-scale convergence is represented by air parcels and, thus, when tracking moisture along the trajectories, the moisture source along different branches of trajectories are identified by WaterSip and ETmixing. If there is substantial (turbulent) mixing of air that leads to changes in the moisture content, such a moisture uptake is identified by WaterSip, thereby losing information on the original surface moisture source. But if the convergence of air is not correctly represented by the trajectories (e.g. because strong turbulence/mixing is involved), also ET-mixing will be misrepresenting the moisture sources. Further, if an air parcel takes up moisture due to subscale processes, also ET-mixing might not identify the original source of this moisture as the trajectory flow does not follow the flow of the moisture in the subgrid processes (e.g. during turbulent mixing). Can you be more specific here to which processes you are referring, and if these processes affect the trajectory calculations or the moisture source identification.

**Reply:** Thank you for your comment. The manuscript text noted above is referring to the moisture source identification. Both methods have advantages and disadvantages. Here we are noting the difference between the methods in terms of their attribution to surface moisture sources; WaterSip has an indirect attribution while ET-mixing has a more direct attribution, due to each method's construction.

The trajectory calculations are the same regardless of the moisture source identification method. In the context of an atmospheric river such as the Scotland case we appreciate that the WaterSip approach well captures the atmospheric convergence associated with the atmospheric river, but differs to ET-mixing in its moisture source identification approach.

To clarify, we have amended line 447 as follows:

"Consequently, the WaterSip method should well capture the convergence-related atmospheric river, but may not necessarily identify the original, surface evaporative sources of the atmospheric moisture."

**Line 481:** The recommendation for Option 4 is only based on theoretical assumption. There is no evidence from the studies results that the combination of ET-mixing and WaterSip correctly represent the theoretical framework and the processes in reality. I would not make any recommendation based on these results.

**Reply:** Based on the available evidence, including the underpinning assumptions, we consider that the ET-mixing provides the best attribution of evaporative moisture source while WaterSip provides the best estimate of moisture uptake when the vertically well-mixed assumption is not valid.

This reasoning for our recommendation will be further clarified in the revised manuscript.

**Line 540:** "underestimate" -> without knowing the true state, I would not used the terms under- or overestimation

**Reply:** Yes, the sentence here is misleading, as there is no "truth" to validate the results. This will be changed to refer to the methods being higher or lower than each other.

**Lines 542-543:** "In comparison, the ET-mixing method more directly identifies regions where moisture originates via evapotranspiration from land or ocean surfaces..." This direct connection to the surface is by construction. The moisture source regions could still be misrepresented (see comment on Lines 447-450).

**Reply:** Thank you, yes it is by construction. Limitations of this method are discussed in the paper e.g. section 3.4.

**Lines 548-550:** "Finally, for the moisture identification, air column dividing and convection are both considered." Does this refer to the recommendation for Option 4? I would not include a recommendation without have a true state for comparison.

**Reply:** Yes, this refers to the recommendation of option4. Recommendations are based on our assessment of the available evidence. However, we will also add that more research is required to analyse the vertical atmospheric mixing in greater detail, because ET-mixing construct the connection between ET and moisture source through the well-mixed assumption.